# ConcatPlexer : Additional Dim1 Batching for Faster ViTs

## Abstract

Transformers have demonstrated tremendous success not only in the natural language processing (NLP) domain but also the field of computer vision, igniting various creative approaches and applications. Yet, the superior performance and modeling flexibility of transformers came with a severe increase in computation costs, and hence several works have proposed methods to reduce this burden. Inspired by a cost-cutting method originally proposed for language models, Data Multiplexing (DataMUX), we propose a novel approach for efficient visual recognition that employs additional dim1 batching (*i.e.,* concatenation) that greatly improves the throughput with little compromise in the accuracy. We first introduce a naive adaptation of DataMux for vision models, Image Multiplexer, and devise novel components to overcome its weaknesses, rendering our final model, ConcatPlexer, at the sweet spot between inference speed and accuracy. The ConcatPlexer was trained on ImageNet1K and CIFAR100 dataset and it achieved 23.5% less GFLOPs than ViT-B/16 with 69.5% and 83.4% validation accuracy, respectively.

## 1 Introduction

Deep learning research community has experienced dazzling advances in model performance across a wide variety of domains and downstream tasks in the last decade [1, 2, 3, 4, 5, 6, 7]. These improvements, however, came at the cost of rapidly increasing computational burden, with the introduction of Transformer [2, 3, 8] marking a major milestone in this aspect. With the growing popularity of transformers, methods to reduce their computational costs have become a prominent research topic [9, 10, 11, 12, 13, 14]. However, previous efforts to improve the computational efficiency of transformers have been mostly focused on the NLP domain. Data multiplexing (DataMUX) [15] pioneered this direction of research for language models by projecting multiple input tokens into a single compact representation space and thus enabling the neural network to process them simultaneously. Although DataMUX [15] has delivered promising preliminary results for the concept of data multiplexing, there is much room for research remaining unexplored especially in the vision domain. For instance, it has mainly trained the transformer on the GLUE benchmark and as a CV task, the authors have only experimented on the MNIST dataset with light Multi-Layer Perceptron (MLP) and Convolutional Neural Network (CNN). This experimental setting is at best a proof-of-concept and thus insufficient to ensure its general applicability in the vision domain.

In this paper, we explore the potential of data multiplexing in larger scale general vision applications such as ImageNet1K [16] classification. To that end, we first show the limitations of naive adaptation of DataMUX by constructing a simple baseline named Image Multiplexer that employs DataMUX for visual recognition with minimal modifications. We then progressively transform this architecture to reach a favorable trade-off between the accuracy and inference speed, presenting our final model, ConcatPlexer. ConcatPlexer, in short, is a method for efficiently extracting multiple images' represen-

Submitted to the Workshop on Advancing Neural Network Training at 37th Conference on Neural Information Processing Systems (WANT@NeurIPS 2023). Do not distribute.

tation at once. ConcatPlexer extracts high-level feature tokens via Transformer encoder layers. Then instead of projecting multiple inputs to a compact representation space, our ConcatPlexer reduces the length of input tokens using a learned convolution and concatenates them for simultaneous processing. Comparison with the naive Image Multiplexer clearly demonstrates that DataMUX in its native form is ill-suited for vision models but can be made effective with our proposed modifications.

The ConcatPlexer and its MultiPlexer baseline are pretrained and compared on ImageNet1K [16], making them generally applicable vision frameworks. We further finetune them on CIFAR100 [17] and evaluate the result. For evaluation of this new framework, we suggest a *"multiplexed image classification task"*, whose goal is to classify multiple images multiplexed into a single representation. ConcatPlexer achieves consistent gains over its baselines in both ImageNet and CIFAR100, supporting its effectiveness in visual recognition tasks.

The contribution of this paper is as follows:

1. This paper defines the "multiplexed image classification task" and deal with the concept of data multiplexing that projects multiple inputs into a single representation for efficient data processing in the vision domain.
2. We propose the ConcatPlexer, a novel framework for multiplexing images, and test its performance on ImageNet1K and CIFAR100 benchmark. The ConcatPlexer extracts high-level featured tokens using the transformer encoder patchifier and concatenates multiple images to process them at once.
3. We demonstrate that data multiplexing can obtain a favorable trade-off between throughput and accuracy. Our model can save up to 66.9% of FLOPs compared to ViT-B/16 with mild drop in accuracy.

## 2   Related Work

**Data Multiplexing:** The concept of data multiplexing was first suggested by DataMUX [15]. The DataMUX processes multiple inputs by projecting multiple texts into a single compact representation space. This enables models to process much larger batch with a same GPU resource. DataMUX and its latter version MUX-PLM [18] demonstrate their performance on GLUE Benchmark [19]. In this work, our proposed ConcatPlexer gains model efficiency by transplanting the data multiplexing method into the vision domain successfully.

**Token Reduction:** Though we are the first to apply the concept of data multiplexing to vision domain to the best of our knowledge, there are studies that try to cut the computational cost of transformer-based models by reducing the number of input tokens. ToMe [14] merges similar tokens to reduce the length of the input sequence. The other approaches [13, 12] prune tokens into a single token to reduce the length of an input sequence. However, our method processes multiple inputs at the same time, naturally reducing the computational cost.

## 3   Method

### 3.1   Preliminary: DataMUX

**Multiplexing:** To project multiple inputs into a single compact representation space, a multiplexing module is used. Consider $(x^1, \cdots, x^N)$ with $x^i \in R^d$ being a tuple of $N$ inputs within a batch. Multiplexing transforms each input by $\phi^i : R^d \mapsto R^d$ and averages at the end. A backbone takes a batch $(x^1, \cdots, x^N)$ as an input and outputs the multiplexed hidden representation output $h^{1:N}$.

$$h^{1:N} = \Phi(x^1, \cdots, x^N) = \frac{1}{N} \sum_{n=1}^{N} \phi^i(x^i). \tag{1}$$

Considering the case for a sequenced token input with length $L$, the aforementioned multiplexing process is done in a token-wise order. For an input sequence $x^i = \{w_j^i\}_{j \in [L]}$, each token $w_j$ can be processed as

$$h_j{}^{1:N} = \Phi(w_j^1, \cdots, w_j^N). \tag{2}$$

In DataMUX, either (1) a random fixed orthogonal matrix or (2) a fixed Gaussian random matrix is used for the linear projector $\phi^i$. In our naively implemented Image Multiplexer, a fixed orthogonal matrix is used.

**Demultiplexing:** To disentangle the multiplexed hidden representation output $h^{1:N}$ into $N$ independent representation, demuxing module is used. Demultiplexing function $\Theta^i$ extracts $i^{th}$ representation from a multiplexed hidden representation as

$$y^i = \Theta^i(h^{1:N}), \forall i \in [N]. \tag{3}$$

For demultiplexing function, there are two choices: (1) MLP Demuxing: using $N$ MLPs to extract $N$ representation from $h^{1:N}$ and (2) Index Embedding. An MLP Demuxing is used for our naively implemented Image Multiplexer.

**Theoretical claim of DataMUX:** The major factor that Transformer based models can handle the multiplexed task is Transformer's multi-head attention mechanism. Each multi-head attention can extract features of different inputs within the multiplexed representation. Kindly refer to DataMUX [15] for more detailed theoretical claim.

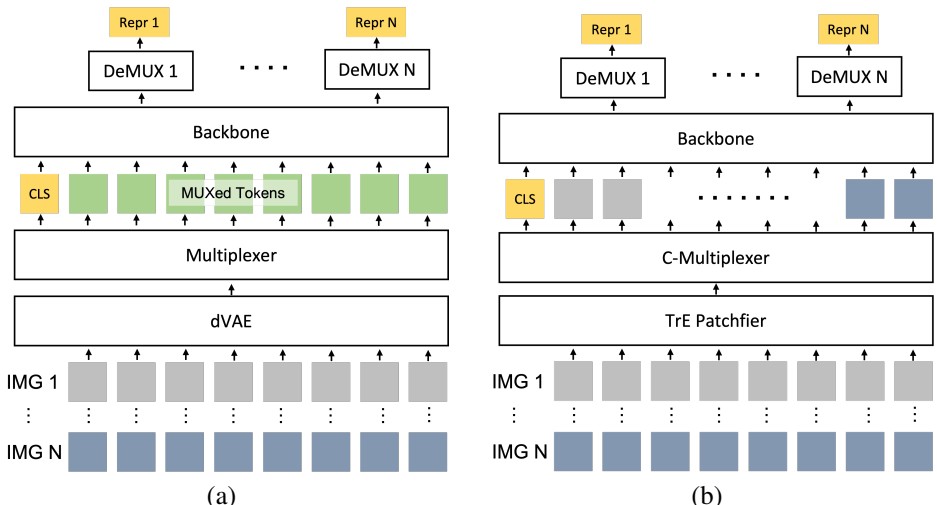

Figure 1: Overall architecture of (a) Image Multiplexer and (b) ConcatPlexer. The Image Multiplexer multiplexes $N_{MUX}$ images using MLP and fixed orthogonal matrices. The ConcatPlexer uses a conv layer to reduce the length of each image token and concatenates them. $N_{MUX}$ is abbreviated as N in this figure.

## 3.2 ConcatPlexer

We propose the ConcatPlexer (Figure. 1-b), a model that successfully adapts the concept of DataMUX to the vision domain by addressing the structural differences between the two modalities. As explained in [20], the most decisive difference between visual signal and natural language lies in data redundancy. A pixel that constitutes an image rarely carries significant information by itself while a word token more likely carries important semantic information. In order to compensate for this difference and suit DataMUX for vision tasks, we compose our ConcatPlexer with the following architectural components: Transformer patchifier, ConcatMultiplexer, Demultiplexer, and the Backbone.

**Transformer Encoder Patchifier:** To address the redundancy issue of pixel-based data, we extract high-level features before feeding to the multiplexing backbone. High-level featured tokens will reduce redundancy. This will make the backbone process and distinguish the multiple inputs parallelly. Transformer Encoder(TrE) Patchifier is a stacked transformer encoder layer with CNN layer at the front. Suppose that the dimension of input images is $(bs, 3, W, H)$ where $bs$ is the batch size, 3 is a color channel, $W$ is the width of an image, and $H$ is the height of an image. First, the CNN layer patchifies the image into a grid patch turning the input dimension into $(bs, L, dim)$ where $bs$ is the

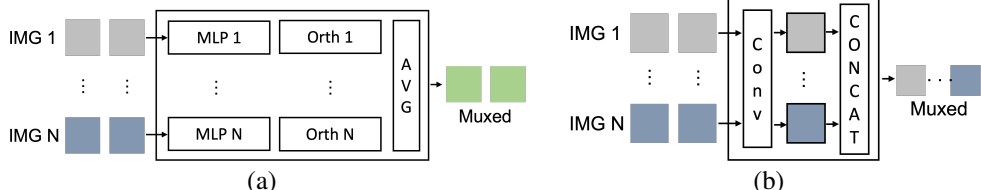

Figure 2: The architecture of (a) Multiplexer and (b) C-Multiplexer. Both inputs $N_{MUX}$ of inputs and combine them into a single input. $N_{MUX}$ is N for this figure.

batch size, $L$ is the token length of each image, and $dim$ is the dimension of each token. Then rear TrE will turn the $(bs, L, dim)$ tokens into a high-level featured token while retaining the size of input the same.

**C-Multiplexer:** While training the Image Multiplexer, we observed that the existing multiplexing method's performance degradation outweighs the efficiency gain for the tasks that have low expecta-tions on random chance. We optimize the trade-off between computational benefit and performance degradation by preventing severe performance degradation while retaining computational efficiency to a certain degree. Instead of projecting multiple($N_{MUX}$) image tokens into a single compact representation space, we tried to extract the essence of each image and combine them in a different manner. In the Figure. 2-b, to extract the essence of each image, conv computation was used on high-level featured tokens. Using the conv1d layer with the output channel $dim$, the dimension of $(bs, L, dim)$ tokens from TrE tokenizer becomes $(bs, L/N_{MUX}, dim)$ where $N_{MUX}$ is number of sample to multiplex. From this, each image gets shorter in length while retaining necessary information as much as possible. Then we concatenated the tokens of $N_{MUX}$ images to train the backbone to process $N_{MUX}$ images at the same time and store $N_{MUX}$ representation in a single CLS token. From this operation, the input of dimension $(bs, L/N_{MUX}, dim)$ becomes $(bs/N_{MUX}, L, dim)$ thereby enables the model to process $N_{MUX}$ times larger batch. As C-Multiplexer is very simple, the computational overhead is negligible.

**Demultiplexer and backbone:** For the Demultiplexer and backbone, the ConcatPlexer uses the same Demultiplexer and backbone as the Image Multiplexer. The backbone is a ViT-like architecture that stacks the transformer encoder layers. The backbone takes $(bs/N_{MUX}, L, dim)$ dimension tokens as an input and outputs $(bs/N_{MUX}, L + 1, dim)$ tokens including the CLS token. $N_{MUX}$ of MLPs were initialized to separate the representation of each image from a single CLS token of the backbone. For more detail, refer to Sec. 4.1.

## 3.3 Training

To train ConcatPlexer, three loss terms were used. Firstly, classification loss using ground truth class label was used. To boost the performance of ConcatPlexer, CLIP loss and Label smoothing loss were used.

**CLIP Loss:** The CLIP loss is intended to train the ConcatPlexer's demultiplexed CLS output to resemble the representation of CLIP vision encoder. By encouraging the model to learn the general representation space of the CLIP encoder, CLIP loss can prevent the model from overfitting and blindly memorizing the ground truth (GT) label. The $CLS_x$ in Eq. 4 is demultiplexed CLS token of an image $x$. $CLIP(\cdot)$ is a CLIP vision encoder that outputs feature token of image $x$. The similarity between the two features is calculated by the contrastive loss.

$$\mathcal{L}_{CLIP} = Ctrs(CLIP(x), CLS_x). \tag{4}$$

**Label Smoothing Loss:** In order to take advantage of the ConcatPlexer's multiplexed input, we augmented the image by mixing other high-level image tokens within the multiplexed sample at

C-Multiplexer. Tokens of $N_{MUX}$ images are averaged as follows:

$$M_i = \sum_{n=1}^{N_{MUX}} f(T_n), \quad f(T_n) = \begin{cases} \alpha * T_n, & \text{if } n = i \\ \frac{(1-\alpha)}{N_{MUX}-1} * T_n, & \text{otherwise} \end{cases}$$

$$M_i^{GT} = \sum_{n=1}^{N_{MUX}} g(Y_n), \quad g(Y_n) = \begin{cases} \alpha * Y_n, & \text{if } n = i \\ \frac{(1-\alpha)}{N_{MUX}-1} * Y_n, & \text{otherwise} \end{cases} \quad (5)$$

$$\mathcal{L}_{smooth} = CE(M_i, M_i^{GT}),$$

where $T_n$ and $Y_n$ are a $n^{th}$ high-level featured tokens among $N_{MUX}$ images after TrE tokenizer and ground truth class label of a $image_n$ in one-hot vector format, respectively. As a result, $M_i^{GT}$ is a smoothed one-hot vector label, whose $i$-th component is set to $\alpha$, and $(1 - \alpha)$ is distributed to other components corresponding to the labels of other images. This prevents the model from overfitting the training dataset and shows a slight performance gain.

# 4 Experiment

## 4.1 Baseline: Image Multiplexer

Along with ConcatPlexer, we introduce Image Multiplexer (Figure. 1-a), a naively implemented version of DataMUX in the vision domain, as a baseline. Unlike DataMUX in NLP, the Image Multiplexer has a long way to go due to several structural differences in vision. For the training, classification loss and token retrieval loss were used. For the token retrieval loss, the model is trained to restore the original discrete input tokens. This helped boost the performance of the original DataMUX. Implementation detail is described in the following section.

**Image Multiplexer:** To bring the DataMUX into a vision regime, Image Multiplexer can be broken down into four parts: (1) Discrete patchifier, (2) Multiplexer, (3) Backbone, and (4) Demultiplexer.

**Discrete Patchifier:** NLP inputs are tokenized into discrete tokens. Original DataMUX exploits this nature with token retrieval task. The discrete patchifier is used to make pixel patched into discrete tokens. Specifically, DALL-E's pretrained discrete variational autoencoder (dVAE) [21] was used. DALL-E's dVAE patchifies 8x8 pixels into a single discrete 13-bit code (total 8192 codes). This enables the model to be trained with token retrieval loss.

**Multiplexer and Demultiplexer:** The Multiplexer multiplexed (Figure. 2-a) $N_{MUX}$ of discretized images into a single muxed input. For the Multiplexing module, $N_{MUX}$ of shallow MLPs and random fixed orthogonal matrices were used. Each discretized image is projected with a shallow MLP and an orthogonal matrix. Then $N_{MUX}$ of projected representations are averaged to be multiplexed into a single compact representation space. For the Demultiplexing module, $N_{MUX}$ of shallow MLPs were used. Each MLP is trained to extract the representation of each image from muxed representation output of the backbone.

The structure of the Multiplexer is the major difference between Image Multiplexer and the Concat-Plexer. Image Multiplexer projects and combines via linear projection and fixed orthogonal matrices while C-Multiplexer uses conv computation to reduce tokens of $N_{MUX}$ images and concatenates them in a single sequence.

**Backbone:** The ViT-like architecture was used for the Image Multiplexer backbone. The backbone shares the same configuration as the ViT-base model [3]. 12 transformer encoder layers were stacked and the representation dimension was 768.

## 4.2 Experimental Detail

For a multiplexed image classification task, Image Multiplexer and ConcatPlexer are trained on ImageNet1K and CIFAR100 datasets. Both models were trained using an AdamW optimizer with a learning rate of 1e-4 and weight decay of 0.03 for ImageNet1K dataset. Each model was trained around 50 epochs until it converged on the training set. As shown in table 1 and mentioned before, Image Multiplexer uses DALL-E [21] tokenizer or CNN patchifier, and ConcatPlexer uses a transformer encoder (TrE) as a high-level featured tokenizer. The $N_{MUX}$ on table 1 means the number of samples

| Name | Tokenizer | Token Length | $N_{MUX}$ | Concat Point | Val Acc |
|---|---|---|---|---|---|
| Image Multiplexer | DALL-E | 784 | 2 | - | 54% |
| Image Multiplexer | DALL-E | 784 | 4 | - | 48% |
| Image Multiplexer | CNN | 196 | 4 | - | 26% |
| ConcatPlexer(1) | TrE | 196 | 2 | 1 | 62.3% |
| ConcatPlexer(2) | TrE | 196 | 2 | 3 | 65.3% |
| ConcatPlexer(3) | TrE | 196 | 2 | 6 | 69.5% |
| ConcatPlexer(4) | TrE | 196 | 4 | 1 | 56.7% |

Table 1: Performance of the Multiplexed Models on ImageNet1K.

| Name | Tokenizer | Token Length | $N_{MUX}$ | Concat Point | Val Acc |
|---|---|---|---|---|---|
| Image Multiplexer | DALL-E | 784 | 2 | - | 74% |
| Image Multiplexer | DALL-E | 784 | 4 | - | 70% |
| ConcatPlexer(1) | TrE | 196 | 2 | 1 | 76.1% |
| ConcatPlexer(2) | TrE | 196 | 2 | 3 | 78.6% |
| ConcatPlexer(3) | TrE | 196 | 2 | 6 | 83.4% |

Table 2: Performance of the Multiplexed Models on CIFAR100.

multiplexed in a single sequence. Each model multiplexed from two samples up to four samples at a single sequence. The 'Concat Point' means at which layer the TrE tokenizer is concatenated. In other words, before concat point each sample is processed independently and after the concat point $N_{MUX}$ samples are concatenated and processed at once. We call layers before and after the concat point as projection layers and backbone layers, respectively. The total number of layers, including both the projection layers and the backbone layers, was set to 12. The batch size of the Image Multiplexer and the ConcatPlexer is 512-1024 to fit the size of GPU memory. The Image Multiplexer was trained on 8 A100 GPUs and ConcatPlexer was trained on 4 A100 GPUs, respectively.

### 4.3 Experiment on ImageNet1K

The aforementioned models are pretrained with ImageNet1K and results are reported in Table 1. As shown in Table 1, performance tends to drop as $N_{MUX}$ parameter increases in both Image Multiplexer and ConcatPlexer. This is natural because $N_{MUX}$ being four means that twice more information should be crammed in the same space as $N_{MUX}$ being two. Also, although the total number of projection layers and backbone layers is kept to 12 in total, the thicker projection layer tends to show better performance in the cost of computational efficiency. This is because the projection layer is a layer that comes before muxing and the backbone is a layer that comes after muxing.

Referring to Table 1, Image Multiplexer using CNN tokenizer saturated at validation accuracy of 26%. Replacing the tokenizer with DALL-E has boosted validation performance up to 48%, muxing 4 image inputs. The DALL-E tokenizer enables image patches to be discrete, but its token length prohibitively increases the computation cost, which makes the purpose of multiplexed image classification task pointless. On the contrary, the ConcatPlexer shows validation accuracy of 56% with muxing four images at the same time using a single layer TrE tokenizer. The ConcatPlexer with $N_{MUX} = 2$ shows validation accuracy of 62% - 69%. Performance gets better as projection Layers increase in the cost of computational efficiency. The ConcatPlexer is compared with conventional ViT[3] most similar backbone but trained with a non-multiplexed image classification task at section 4.5.

### 4.4 Experiment on CIFAR100

The pretrained models from Table 1 were finetuned on the CIFAR100 dataset. Similar to Table 1, the ConcatPlexer with smaller Num Muxed and larger projection Layers performs better in the cost of computational cost. Referring to the Table 2, the performance of the ConcatPlexers outperforms the Image Multiplexers with less computational cost. As the model is trained on easier task (smaller number of classes), the degradation gap between ConcatPlexer and ViT reduces. According to Table 2 and Table 3, ConcatPlexer(3)'s validation accuracy is 83.4% and ViT-B/16's validation accuracy is 87.13%.

| Model | GFLOPs |
|---|---|
| ViT-B/16 | 17.58 |
| ConcatPlexer(1) | 9.81 |
| ConcatPlexer(2) | 11.26 |
| ConcatPlexer(3) | 13.45 |
| ConcatPlexer(4) | 5.81 |

Table 3: FLOPs Count Per Image.

| Model | Dataset | Val Acc |
|---|---|---|
| ViT-B/16 | INET1K | 77.91% |
| ConcatPlexer(3) | INET1K | 69.5% |
| ViT-B/16 | CIFAR100 | 87.13% |
| ConcatPlexer(3) | CIFAR100 | 83.4% |

Table 4: Comparison with ViT.

| Model | ImageNet1K Val Acc |
|---|---|
| W/O MUX | 60.7% |
| MUX | 62.3% |

Table 5: Thicker Batch vs. Multiplexing.

| IR | @1 | @5 | @10 |
|---|---|---|---|
| MMP | 32.84% | 59.62% | 71.42% |

Table 6: Flickr zero-shot image retrieval.

## 4.5 Ablation

**Comparison with non-multiplexing method:** The ConcatPlexer uses TrE patchifier to get high-level patch tokens. Then token length is reduced by the conv layer and multiple inputs are concatenated. Instead of concatenating and just stacking the reduced length inputs after conv computation may look like a good option. Table 5 indicates that ConcatPlexer performs better than *Without MUX model*.

**Comparing with conventional ViT:** Table 4 indicates that ConcatPlexer lacks performance in ImageNet1K compared to ViT-B/16 . This is because the ConcatPlexer is tackling a harder task: multiplexed image classification. However, the performance gap narrows if the model is trained on an easier dataset: CIFAR100. Considering that we are the first to propose the multiplexed image classification task, there is more room for improvement. Therefore, we believe that the current aspect seems encouraging.

The computational cost of the Image Multiplexer using CNN patchifier was not calculated as its performance was not comparable with other models. The computational cost of the Image Multiplexer using DALL-E dVAE was also not calculated as its computational cost was prohibitively large due to the long sequence length.

As the main purpose of the ConcatPlexer is to attain increased computation efficiency and throughput, we also compared the FLOPs of the ConcatPlexers and ViT-B/16. Table 3 indicates that the Concat-Plexers require less FLOPs compared to ViT. The FLOPs were counted using fvcore library of Meta Research.

**Comparison with original DataMUX:** Original DataMUX and its descendent [15, 18] demonstrated its effectiveness on GLUE benchmark [19]. However, an expectation of random chance of CIFAR100 and ImageNet1K is much lower, considering that tasks in the GLUE benchmark usually have two to three classes to predict. Of course, DataMUX shows an impressive performance in the aspect that it can multiplex extremely many inputs (up to 10 for MUX-PLM). The ConcatPlexer multiplexes fewer inputs and the performance gap may be seen as quite large. However, the ConcatPlexer deals with the trickier task in the sense of expectation of random chance.

**Possibility toward multimodal multiplexing:** As a proof of concept, we adapt data multiplexing in the Vision&Language(VL) domain: Multimodal Multiplexer (MMP). It is a single ViT-architectured model that represents images and texts in a modality-agnostic manner, with no modality-specific transformer blocks or projection layers. As shown in Table 6, the model achieves 32.84%, 59.62%, and 71.42% accuracy at recall@1, recall@5, and recall@10 on Flickr30K [22] with zero-shot image retrieval task. Despite its modality-agnostic information processing mechanism and parameter efficiency (compared to typical VL models that employ separate transformer towers for each modality), MMP achieves promising preliminary results on challenging multimodal retrieval task, which we believe indicates great potential for future follow-up works.

## 5    Limitations and Future works

We propose a new vision classification called multiplexed image classification task to multiply the throughput of the neural network. Though this approach can drastically drag up the computational efficiency and throughput by increasing $N_{MUX}$, the concept of calculating multiple data at the same time makes the previous task much harder, which causes a clear trade-off between $N_{MUX}$ and performance.

The performance of transformer-based models in vision tends to be heavily affected by hyperparameter tuning. With more delicate tuning, the ConcatPlexer may have more performance gain. As our Conv-based multiplexing method is somewhat heuristic, we believe that there is more room for improvement if other token length reduction methods are used together. Also, there is room for computation efficiency gain of the ConcatPlexer if we use a method similar to Swin Transformer [23], which separates images from the entire sequence and divides them into partitions, applies local self-attention, and combines information from the CLS token.

## 6    Conclusion

From this paper, we would like to bring constructive discussion for the computational efficiency of transformer-based models. Our paper tries to transplant the idea of DataMUX [15] of NLP into the vision field. We propose the multiplexed image classification task and its baseline ConcatPlexer and Image Multiplexer. The proposed model shows the feasibility that idea of DataMUX can be applied in the vision field.

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
