# OpenReview forum: "ConcatPlexer : Additional Dim1 Batching for Faster ViTs"
_NeurIPS.cc/2023/Workshop/WANT — WANT@NeurIPS 2023 Oral_

### Official Review · Reviewer_4B4m · 2023-10-19
**The paper presents a new cost-cutting approach for ViTs that uses additional concatenation batch processing, which improves computational efficiency while trading off accuracy. A multiplexed image classification task is proposed to classify multiple images multiplexed into a single representation.**

**Rating:** 5
**Confidence:** 3

**Review:**

Pros:
The proposed method is the first data multiplexing approach that extends DataMUX, a cost-cutting method for language models, to vision transformer models.
The experiments and the ablation study are carefully designed, the results are clearly presented.

Cons:
The trade-off between performance gain at the expense of accuracy is not that significant. There seems to be something important missing when adapting DataMUX to the image classification task.
The best performing model, ConcatPlexer(3), multiplexes up to 4 inputs and requires 13.45 FLOPS compared to 17.58 for ViT-B/16, but the accuracy drops by 8.41% and 3.73% on the INET1K and CIFAR100 datasets, respectively. In comparison, DataMUX multiplexes up to 40 inputs, achieving up to 11x/18x increase in inference throughput with absolute performance drops of < 2% and < 4%.
The text is not always consistent:
1. There is a mix of present and past tenses on lines 114-134.
2. Table 5 is mentioned before Table 4, but is numbered higher.

---

### Official Review · Reviewer_2tTQ · 2023-10-26
**Good Paper, Important Topic**

**Confidence:** 3

**Review:**

# Conclusion

## Cons

* Authors didn't provide any ablation of using CLIP and label smoothing losses neither in their setup nor in reimplementation of the baseline. Also it wasn't clear to me why not to try/ablate DALL-E and CNN patchifiers with ConcatPlexer

## Pros

* Authors perform very reasonable and meaningful research on the topic
* Evaluation shows some performance improvements
* The presentation is very clear and easy to follow

This paper develops around very interesting idea - that it's possible to utilize the multidimensional structure and/or spatial scale of objects' space in order to "embed" several data points from a batch into the original space and hence improve model's throughput. Authors concentrate on computer vision tasks and propose to use convolution followed by concatenation instead of matrix multiplication followed by averaging, this aligns with the intuition that in images' domain spatial correlation between neighbor pixels can help with reduction of the dimension. I think the proposed method is an important topic for the further research.

# Remarks

## Architecture

* It would be interesting to see if authors have tried to use several convolutional layers with non-linearities, likewise in the baseline model's MLP for multiplexing step

* In some scenarios the proposed algorithm ConcatPlexer can be considered as a special case of the original DataMUX, and hence it's unclear if DataMUX can or cannot learn the same representation as the proposed method. For example, consider a multiplexing of 2 elements $x_{1, 2} \in \mathbb{R}^d$ then $\Phi_{\text{ConcatPlexer}}(x) = [A x_1, A x_2]$, where $A \in \mathbb{R}^{d \times (d / 2)}$ is a matrix that represents convolution operation, clearly there exist block matrices matrices $B_{1,2} \in \mathbb{R}^{d \times d}$ such that
$B_1 = 2 [ A_1, 0], \quad B_2 = 2 [ 0, A_2]$. And then
$$\Phi_{\text{DataMUX}}(x) = \frac{1}{2} (B_1 x_1 + B_2 x_2) = [A x_1, 0] + [0, A x_2] = [A x_1, A x_2] = \Phi_{\text{ConcatPlexer}}(x)$$
If I understood correctly it would be possible to get such result by some regularization or constraining of matrices from DataMUX

## Experiments

* It's quite unclear why CNN tokenizer gives such a bad result on validation

---

### Meta-Review · Area_Chair_e8RC · 2023-10-26

**Recommendation:** Accept (Poster)
**Confidence:** 4

**Metareview:**

The paper introduces a new, interesting technique aimed at improving the efficiency of vision transformers. Inspired by DataMultiplexing (DataMUX), originally introduced in NLP, the paper employs additional dim1 batching (i.e., concatenation), resulting in faster inference with little to no drop in accuracy.

Reviewer 2tTQ is quite positive about the idea, while Reviewer 4B4m has some concerns regarding the efficiency gains. Overall, put in balance I believe the idea and approach is interesting enough to be considered for publication.

---

### Decision · Program_Chairs · 2023-10-28

**Decision:**

Accept (Oral)

**Comment:**

We thank the authors for their time and contribution to WANT and we are pleased to share that after the reviewing process the paper has been accepted. Congratulations! We encourage the authors to consider reviewers' feedback for the improvement of the camera-ready version. We hope to see you in person at the workshop and brainstorm on efficient training research together!